# Evaluation of pharmacokinetics, safety, and efficacy of [211At] meta-astatobenzylguanidine ([211At] MABG) in patients with pheochromocytoma or paraganglioma (PPGL): A study protocol

**Masao Kobayakawa** [1]*, **Tohru Shiga**[2], **Kazuhiro Takahashi**[2], **Shigeyasu Sugawara**[2], **Kaori Nomura** [2], **Kazuhiko Hanada**[3], **Naoki Ishizuka**[4], **Hiroshi Ito**[2]

1 Medical Research Center, Fukushima Medical University, Fukushima, Japan, 2 Advanced Research Center, Fukushima Global Medical Science Center, Fukushima Medical University, Fukushima, Japan, 3 Department of Pharmacometrics and Pharmacokinetics, Meiji Pharmaceutical University, Kiyose, Tokyo, Japan, 4 Center for Digital Transformation of health, Graduate School of Medicine, Kyoto University, Kyoto, Japan

* mkobaya@fmu.ac.jp

**Data Availability Statement:** No datasets were generated or analysed during the current study. All

## Abstract

### Background

Pheochromocytoma, or paraganglioma (PPGL), is a tumor that arises from catecholamine-producing chromaffin cells of the adrenal medulla or paraganglion. Systemic therapy, such as the combination of cyclophosphamide, vincristine, and dacarbazine or therapeutic radio-pharmaceuticals such as [131I] meta-iodobenzylguanidine (MIBG), may be administered in cases of locally advanced tumors or distant metastases. However, the current therapies are limited in terms of efficacy and implementation. [211At] meta-astatobenzylguanidine (MABG) is an alpha-emitting radionuclide-labeled ligand that has demonstrated remarkable tumor-reducing effects in preclinical studies, and is expected to have a high therapeutic effect on pheochromocytoma cells.

### Methods

We are currently conducting an investigator-initiated first-in-human clinical trial to evaluate the pharmacokinetics, safety, and efficacy of [211At] MABG. Patients with locally unresect-able or metastatic PPGL refractory to standard therapy and scintigraphically positive [123I] MIBG aggregation are being recruited, and a 3 + 3 dose escalation design was adopted. The initial dose of [211At] MABG is 0.65 MBq/kg, with a dose escalation in a 1:2:4 ratio in each cohort. Dose-limiting toxicity is observed for 6 weeks after a single bolus dose of [211At] MABG, and the patients are observed for 3 months to explore safety and efficacy profiles. The primary endpoint is dose-limiting toxicity to determine both maximum tolerated and recommended doses. The secondary endpoints include radiopharmacokinetics, urinary radioactive excretion rate, urinary catecholamine response rate, objective response rate,

relevant data from this study will be made available upon study completion.

**Funding:** This study is funded by the Advanced Clinical Research Center management business subsidy from Japanese government and granted by the Japan Agency for Medical Research and Development(23ck0106815h0001). The funders did not and will not have a role in study design, data collection and analysis, decision to publish, or preparation of the manuscript.

**Competing interests:** The authors have declared that no competing interests exist.

progression free survival, [123I] MIBG scintigraphy on reducing tumor accumulation, and quality of life.

## Trials registration

jRCT2021220012 registered on 17 June 2022.

## Introduction

Pheochromocytoma and paraganglioma (PPGL) are rare tumors that arise from catecholamine-producing chromaffin cells of the adrenal medulla or paraganglion. The annual incidence of PPGL ranges from 0.04 to 0.95 cases per 100,000 person-years [1–5]. In the WHO tumor classification of endocrine tumors published in 2017, PPGL was assigned a disease code of malignancy (ICD-3) [6].

Anticancer drug therapy may be performed in cases where resection by surgery is difficult, distant metastasis is identified, or recurrence occurs after surgery. The most commonly reported therapy is the combination of cyclophosphamide, vincristine, and dacarbazine (CVD) [7,8]. It has been reported that CVD therapy only reduces tumor size in about half of all cases, and may lead to improvements in quality of life (QOL), such as improved symptoms, in the short- to medium-term. [131I] meta-iodobenzylguanidine (MIBG) may also be administered in cases of locally advanced tumors or distant metastases [9,10]. Since September 2021, [131I] MIBG injection has been approved in Japan to treat "MIBG accumulation-positive unresectable PPGL." However, patients should be quarantined from other people for 5–7 days after administration of the drug, because [131I] I emits β-rays, and its half-life is approximately 8 days. [131I] MIBG also emits γ-rays, making it difficult for medical staff to easily access the patients in an isotope treatment room and respond to situations that require emergency medical intervention.

Astatine emits radiation and decays into another element. Among its isotopes, [211At] At is a nuclide that emits α-rays and decays to the stable element lead-207 ($^{207}$Pb) (half-life is 7.2 h). [211At] meta-astatobenzylguanidine (MABG) is a drug in which [211At] is incorporated into a substance called benzylguanidine, which has a chemical structure similar to that of norepinephrine [11]. [211At] MABG is a substrate for the norepinephrine transporter (NET), which takes up norepinephrine, and is taken up by pheochromocytoma cells via NET. In pheochromocytoma cells, NET expression is enhanced compared to normal cells, and a large amount of [211At] MABG is taken into the cells. Because [211At] MABG is stored in pheochromocytoma cells, we hypothesize that it could be a new treatment option for malignant PPGL.

Reductions in tumor size and amount of catecholamine secretion have been reported by non-clinical studies of this drug [12–17] and clinical trial results of [131I] MIBG [9,10], a similar drug that emits β-rays. We conducted a toxicology study required prior to the first dose in human in accordance with the International Comprehensive Harmonization (ICH) -M3 guidelines "Guidance on Nonclinical Safety Studies for The Conduct of Human Clinical Trials and Marketing Authorization for Pharmaceuticals" and ICH-S9 guidelines "Nonclinical Evaluation for Anticancer Pharmaceuticals" (data not published). The present study is an investigator-initiated first-in-human clinical trial in patients with PPGL who have no other appropriate therapy, that has been conducted from September 2022 until present.

## Materials and methods

### Study purpose

The primary purpose of the present study is to determine the Maximum Tolerated Dose (MTD) and Recommended Dose (RD) of [²¹¹At] MABG by assessing the Dose-Limiting Toxicity (DLT), when administered to patients with PPGL. We also examine radiopharmacokinetics, safety profiles, and exploratory efficacy.

### Type of trial

Type of study: Exploratory study.

Clinical Trial Phase: Phase I.

### Study design

This is a single-arm, open-label trial evaluating the tolerability of a single bolus intravenous dose of [²¹¹At] MABG in patients with PPGL. The purpose is to determine the MTD and RD of this drug. Three doses are administered in separate cohorts; Cohort 1 (0.65 MBq/kg), Cohort 2 (1.3 MBq/kg), and Cohort 3 (2.6 MBq/kg), starting with Cohort 1 and investigating RD according to the modified Fibonacci dose escalation method (3 + 3 design).

### Trial period

The trial started in September 2022, and we are currently recruiting patients at Fukushima Medical University Hospital.

The trial period for each cohort is from the date of obtaining informed consent to 12 weeks after administration. The trial schedule for each patient is shown in Fig 1.

### Investigational drug

Investigational drugs (Fig 2) are manufactured in-house by the Advanced Clinical Research Center, in the Fukushima Global Medical Science Center at Fukushima Medical University. Details and handling of the investigational drug are specified in the investigator's brochure and the "Procedures for the Management of the Investigational Drug" separately.

### Definition of dosage and administration

The patients receive a single bolus intravenous dose of the study drug. The dose and dose escalation schedule for each cohort are shown in Table 1.

The dose escalation follows a common 3 + 3 design. Starting with Cohort 1, the planned number of cases in each cohort is three to six, and the addition of cases is stopped when two cases of DLT occur. Fig 3 shows the dose escalation method. If the number of cases falls below 2/3 or 2/6 at the dose level, up to six additional cases are examined at one level lower.

### Definition of DLT

The evaluation period is from study drug administration to Day 43. No further subjects are enrolled until the DLT adjudication period is over. DLT includes Common Terminology Criteria for Adverse Events v5.0 (CTCAE v5.0 Japanese translation JCOG/JSCO version) (CTECA) Grade 4 hematological toxicity and Grade ≥ 3 non-hematological toxicity and

| | Enrollment | Baseline | Administration | Follow-up | | | | | | | | |
|---|---|---|---|---|---|---|---|---|---|---|---|---|
| | Outpatient | | | In hospital | | | | | Outpatient/in hospital | | | |
| Time point | Day -42 to -14 | Day -7 to -1 | Day 1 | Day 2 | Day 3 | Week 1 | Week 2 | Week 4 | Week 6 | Week 8 | Week 12 | |
| Informed consent | X | | | | | | | | | | | |
| At-211 MABG administration | | | X | | | | | | | | | |
| KI premedication | | X | X | | | | | | | | | |
| 5-HT₃ RA premedication | | | X | | | | | | | | | |
| ECOG PS | X | X | | X | X | X | X | X | X | X | X | |
| Weight | X | | | | | | X | X | X | X | X | |
| Physical examinations | X | X | X | X | X | X | X | X | X | X | X | |
| Vital sings and Spo2 | X | X | X | X | X | X | X | X | X | X | X | |
| Blood and chemistry tests | X | X | X | X | | X | X | X | X | X | X | |
| Urinalysis | X | X | | | | | | X | | X | | |
| Electrocardiograph | X | | X | X | | X | X | X | X | X | X | |
| Cardiac ultrasonography | X | | | | | | | X | | X | | |
| Catecholamines in urine | X | | | | | | | X | | X | | |
| I-131 MIBG scintigraphy | X | | | | | | | X | | X | | |
| Computed tomography | X | | | | | | | X | | X | | |
| Quality of life | X | | | | | | | X | | X | | |
| Radiopharmacokinetics | | | ←————————→ | | | | | | | | | |
| Urinary radioactivity | | | ←————————→ | | | | | | | | | |
| Adverse events | | | ←——————————————————————————————→ | | | | | | | | | |

**Fig 1. Trial schedule for each patient.** MABG: Meta-astatobenzylguanidine, KI: Potassium iodine, 5-HT₃: 5-hydroxytryptamine, ECOG PS: European comprehensive cancer group performance status, SpO₂: Peripheral capillary oxygen saturation, MIBG: Meta-iodobenzylguanidine.

febrile neutropenia, anemia requiring red blood cell transfusion, and thrombocytopenia requiring platelet transfusion. However, the following adverse events are excluded:

- Loss of appetite or fatigue.

- Grade 3 nausea or vomiting not requiring gavage or TPN, and Grade 3 diarrhea without prolonged hospitalization. However, the event must be manageable to Grade ≤ 2 within 7

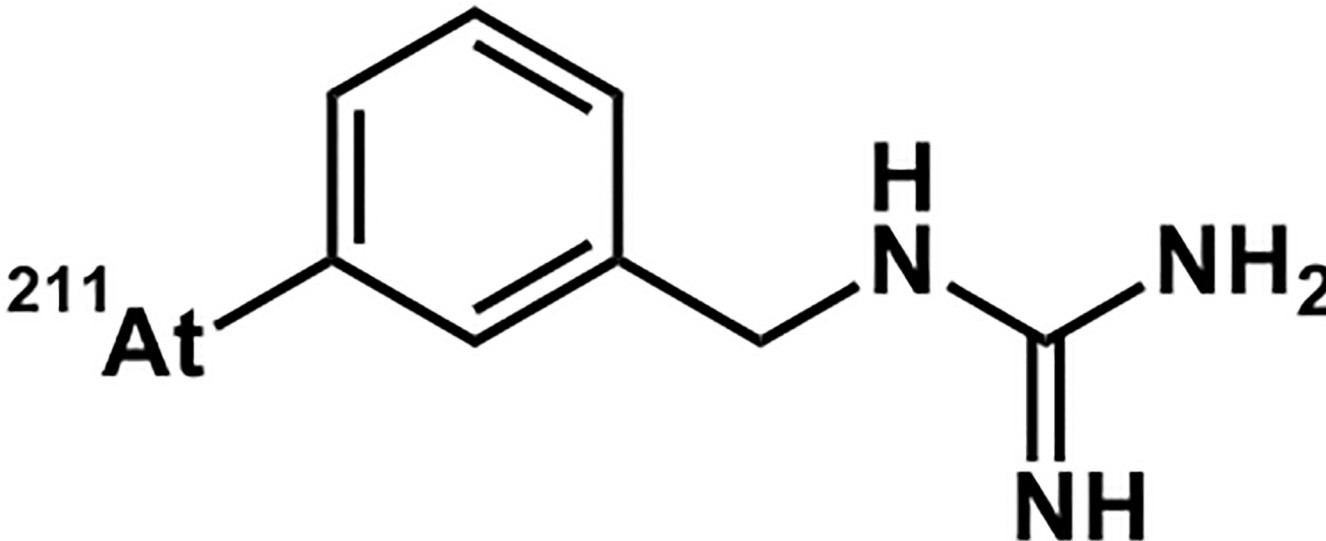

**Fig 2. Structure of [²¹¹At] meta-astatobenzylguanidine (MABG).**

**Table 1. Dosage per level.**

| Cohort | Dose level | Dose (Allowance) | Number of cases |
|---|---|---|---|
| 1 | Level 1 | 0.65 MBq/kg (0.485–0.715 MBq/kg) | 3–6 cases |
| 2 | Level 2 | 1.3 MBq/kg (1.17–1.43 MBq/kg) | 3–6 cases |
| 3 | Level 3 | 2.6 MBq/kg (2.34–2.86 MBq/kg) | 3–6 cases |

days after onset with standard antiemetics or antidiarrheals used at the package insert dosage.

- Grade 3 infection.

The final DLT is determined after consultation between the investigator and sub-investigator for each subject.

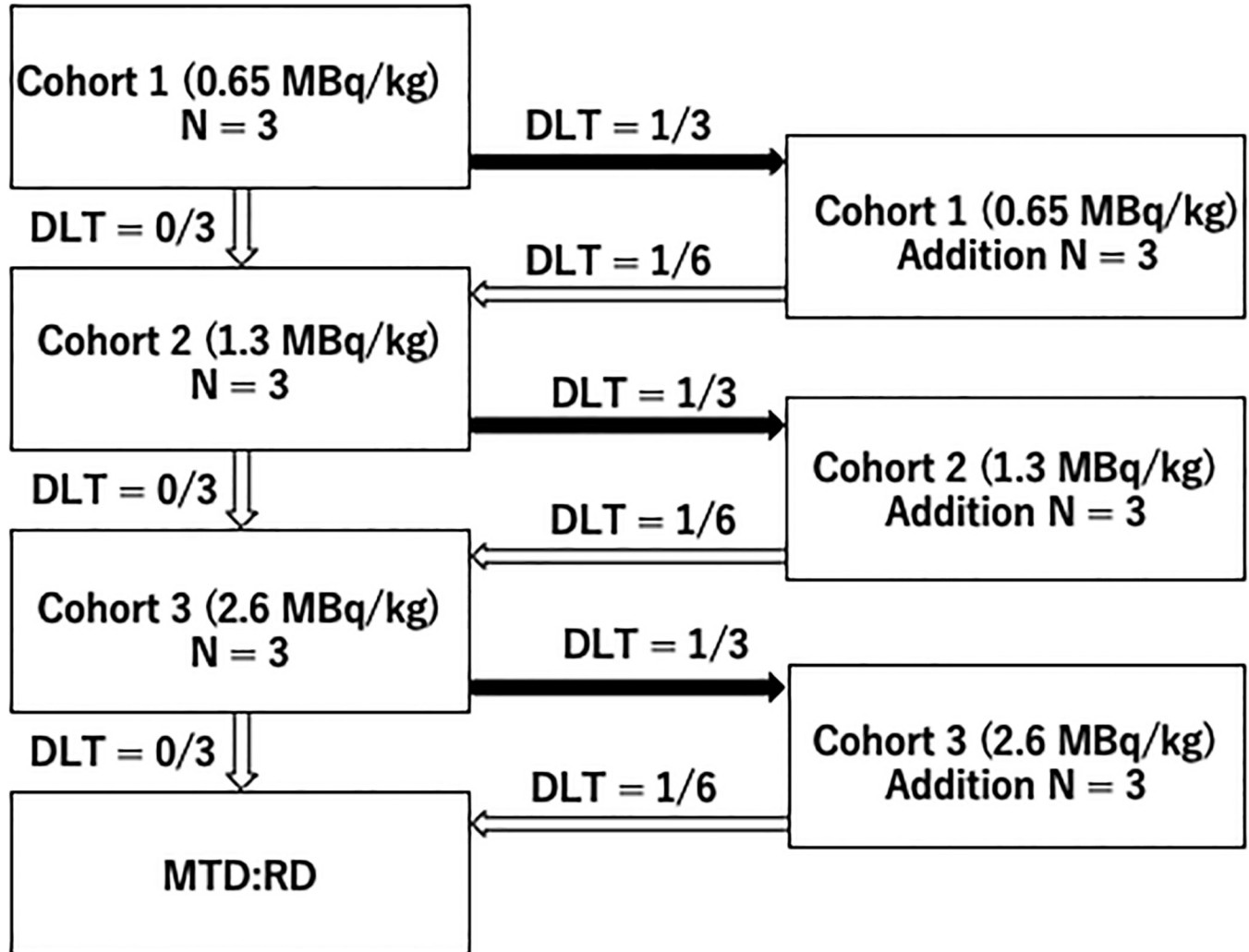

**Fig 3. Dose escalation method.** DLT: Dose limiting toxicity, MTD: Maximum tolerated dose, RD: Recommended dose.

### DLT evaluation procedure in each cohort

1. From the first case of each cohort when DLT appears, the details are reported to the Efficacy and Safety Evaluation Committee (hereinafter referred to as the committee) and the committee holds a meeting.

2. The committee considers the necessity of measures such as discontinuation or interruption of the clinical trial. The investigator refers to the results of the review by the committee and decides on future measures. Furthermore, enrollment of new subjects and administration of the study drug to subjects are interrupted until future measures are decided.

3. At the end of the DLT evaluation period for the third or sixth subject in each cohort, an efficacy/safety evaluation committee convenes to consider the necessity of measures such as discontinuation or interruption of the study. At that time, all available data is confirmed, and the investigator decides whether to proceed to the next cohort, referring to the results of the review by the committee.

### Primary endpoint

The primary endpoint is the presence or absence of DLT to determine the optimal dose for single administration of the study drug.

### Secondary endpoint

1. Radiopharmacokinetics (RPK)
   To measure changes in plasma concentration after administration of an investigational drug to patients, blood is collected according to the following schedule, and the geometric mean of $C_{max}$, AUC, $t_{1/2}$, $V_{ss}$, and CL are calculated as pharmacokinetic parameters in each cohort. The details of the pharmacokinetic analysis are described in the separate pharmacokinetic analysis plan.

2. Urinary radioactivity excretion rate
   Urinary excretion of radioactivity is measured up to 24 h after administration in all cases, and is calculated chronologically in each cohort.

3. Urinary catecholamine response rate
   Patients with urinary catecholamineuria (adrenaline, noradrenaline, metanephrine, and normetanephrine) $\geq$ 3 times the upper limit of normal at screening are eligible. For these patients, the response is determined when the best overall response of these catecholamines achieves CR (decrease in values of all urinary catecholamines subject to evaluation of efficacy to within the standard values) or PR (decrease of > 50% relative to baseline in all urinary catecholamine levels included in the efficacy assessment).

4. Objective response rate (ORR)
   ORR is assessed according to Response Evaluation Criteria in solid tumors (RECIST ver. 1.1) for tumors. In addition, CR and PR are not confirmed at the time of assessment.

5. Progression free survival (PFS)
   After carrying out a primary objective evaluation of the tumor based on RECIST ver. 1.1, this value is defined as the time from the date of enrollment to death or progression (including clinical progression), whichever comes first.

6. Evaluation of decreased [$^{123}$I] MIBG accumulation based on scintigraphy
A comprehensive evaluation of decreased [$^{123}$I] MIBG accumulation based on scintigraphy is performed according to the procedure manual for image evaluation.

7. Quality of life (QOL)
Measure EORTC QLQ-C30 (EORTC-questionnaire Request ID: 83325), EQ-5D-5L. (registration tracking number 49187)

## Sample size

The maximum sample size is 18, according to the modified Fibonacci dose escalation method (traditional 3+ 3 method, up to the third cohort).

## Recruitment and informed consent

Patients are recruited at Fukushima Medical University Hospital. Many patients across Japan are referred to this hospital to participate in the clinical trial. Each participant provides written informed consent before any study procedures are performed.

## Inclusion criteria

All of the following conditions must be met:

1. Patients from whom written consent can be obtained.

2. Patients with histologically or clinically diagnosed pheochromocytoma, paraganglioma, malignant pheochromocytoma, or malignant paraganglioma.

3. Patients diagnosed with pheochromocytoma as defined below (refractory pheochromocytoma is defined as those who satisfy any of the following (a) to (c) and cannot undergo surgical resection or radical external irradiation).

   a. Pheochromocytoma/paraganglioma with an extensive local extension of the primary tumor at first presentation.

   b. Malignant pheochromocytoma/malignant paraganglioma with distant metastasis at first presentation.

   c. Pheochromocytoma/paraganglioma with local recurrence or distant metastasis despite surgical resection.

4. Patients aged $\geq$ 20 years at the time of informed consent.

5. Patients with an ECOG Performance status (PS) of 0–2.

6. [$^{123}$I] MIBG aggregation positive as determined by scintigraphy at the screening in one or more target lesions confirmed by CT imaging.

7. Patients who meet all the following criteria with test values at screening:

   a. bone marrow function.

      1. $\geq$ 3,000/μL without administration of granulocyte colony-stimulating factor (G-CSF).

      2. Hemoglobin without transfusion $\geq$ 9.0 g/dL.

      3. Platelet count without transfusion $\geq 10 \times 10^4$ / mm3 (μL).

 b. Renal function.

 1. Estimated glomerular filtration rate (eGFR) $\geq$ 30 mL/min/1.73 m$^2$.

 c. Liver function (JSCC standardized method).

 1. AST $\leq$ 90 U/L.

 2. ALT $\leq$ 126 U/L (male), ALT $\leq$ 69 U/L (female).

 3. LDH < 666 U/L.

 d. Heart function.

 1. NYHA Functional class: I or lower.

 e. Diabetes.

 1. HbA1c < 8.0% (NGSP value).

 f. Respiratory condition.

 1. Oxygen saturation (SpO2) $\geq$ 96% while breathing ambient air.

8. Patients who are expected to survive for 3 months or longer.

9. Patients who are expected to be independently eat, excrete, and sleep during the nuclear medicine treatment hospital isolation period.

10. Patients for whom standard therapy (CVD therapy or [131I] MIBG therapy) failed or who have no suitable therapy other than that administered in the present study.

## Exclusion criteria

Patients who meet any of the following criteria are excluded.

1. Patients with multiple active cancers.
"Multiple active cancers" refers to the presence of malignancies other than the one included in the study or history of other malignancies with a disease-free interval < 5 years. However, lesions equivalent to carcinoma in situ or carcinoma in situ that are judged to have been cured by local treatment and have a disease-free period of more than 1 year after cure are not included in active multiple cancers. In addition, the following cancers associated with familial pheochromocytoma are not included in active double/multiple cancers:

 a. Medullary thyroid carcinoma in multiple endocrine neoplasia type 2 (MEN2)

 b. Retinal hemangioblastoma in von Hippel Lindau disease (VHL)

 c. Neurofibromas in neurofibromatosis type 1 (NF1)

2. Patients who cannot stop taking a drug that suppresses the accumulation of MABG according to the EANM procedure guidelines for 131I-meta-iodobenzylguanidine (131I-mIBG) therapy [18] during the study period.

3. Patients unable to stop taking α-methylparatyrosine during the study period.

4. Patients who have undergone surgery, CVD therapy, catheter hepatic artery embolization for liver metastasis, or radiotherapy within 8 weeks before enrollment.

5. Patients who have received MIBG treatment within 12 weeks before enrollment.

6. Patients who have developed Grade 2 or higher non-hematological toxicity during or after prior treatment, for which a causal relationship to treatment cannot be ruled out, and who require treatment during the study period.

7. Patients who have any of the following infections and require medical treatment during the study period:

   1. Hepatitis B virus infection

   2. Hepatitis C virus infection

   3. HIV infection

   4. Other infectious diseases requiring systemic treatment

8. Patients with a disease that requires continuous systemic administration of adrenocortical hormone (prednisone or prednisolone equivalent dose of ≥ 10 mg/d) or other immunosuppressants and that requires treatment during the study period.

9. Patients with a history of uncontrolled catecholamine seizures.

10. Patients with a history of fatal arrhythmia or cardiac arrest.

11. Patients with uncontrolled symptomatic arrhythmia, thyroid dysfunction including hypothyroidism and hyperthyroidism, respiratory disease, pleural effusion or ascites.

12. Patients with coronary artery disease, arrhythmia requiring treatment with amiodarone, severe valvular disease, aortic disease, or any disease or condition with bleeding tendency.

13. Patients that are pregnant (even if it is determined that there is a possibility of pregnancy via a doctor's consultation, the patient is excluded from this study), have given birth within the previous 28 days, or are currently breast-feeding (including women who have temporarily stopped breast-feeding).

14. Female patients of childbearing potential or male patients with partners of childbearing potential who are unable to agree to contraception for 6 months after drug treatment (any two of the following: contraception methods that include latex condoms [used by men], oral contraceptives, intrauterine device [IUD] [progesterone-free T type] used in combination; or tubal ligation and vasectomy).

15. Patients participating in other clinical trials within 3 months before the consent date.

16. Patients who are judged by the investigator or sub-investigator to be unsuitable for this study for other reasons.

## Adverse Events (AEs)

An adverse event (AE) is any untoward medical occurrence in a subject, regardless of whether or not it is causally related to the drug, including unintended signs and clinically significant changes in laboratory test values, including exacerbation of disease, symptoms, and complications. New AEs occurring between obtaining informed consent and up to 12 weeks after administration are recorded in the case report form. Discontinuation of study participation due to AEs should also be recorded in the case report form. Worsening PPGL is treated as disease progression under efficacy assessments, not as an AE. All AEs are followed until resolution to baseline status (in terms of Grade), clinical stabilization of symptoms if recovery is unlikely, or until the subject is no longer available for follow-up. When treatment such as

examination or surgery is administered, the treatment itself is not treated as an AE; rather the disease that led to the treatment is considered an AE. Regarding the death of a subject, the disease that causes death is regarded as an AE. If the subject's condition is the same as before treatment and the expected effects of this drug are not observed despite the administration of this drug, it is not regarded as an AE. In the present study, the CTCAE is used as the evaluation criteria for AEs. Any worsening of the severity of each event is treated as an AE and described in the case report form.

## Serious Adverse Events (SAEs)

The definitions of SAEs follow ICH-E2A "Clinical Safety Management: Definitions and Standards for Expedited Reporting" (Oct 27, 1994) and other national regulations. An SAE refers to any unfavorable medical event that occurs in a subject between the date of informed consent and 12 weeks after administration.

## Statistical analysis

This section describes the outline of the statistical analysis plan, and the details are described in the separately specified statistical analysis plan. Sample size is set according to the modified Fibonacci dose escalation (traditional 3 + 3 design). MTD is determined as the highest dose level at which none of the three cases developed DLT or only one of the six cases developed DLT. RD is determined as the same dose of MTD. All secondary endpoints are described as descriptive statistics, no formal statistical hypothesis test is performed. Statistical analyses were performed with SAS software, version 9.4 (SAS Institute). The version of the statistical analysis plan is created when the database is fixed as the final version, and the final analysis is performed according to this version.

As appropriate, summary statistics are calculated for subject demographics, safety, and PK data by cohort or time point. Summary statistics for continuous data include means, medians, standard deviations, and ranges (e.g., $C_{max}$, AUC, geometric mean, and geometric coefficient of variation for PK parameters), and frequencies and proportions are calculated for categorical data. The data is also illustrated as necessary. Changes from baseline to post-treatment or post-treatment percent change relative to baseline are performed in subjects with baseline and post-treatment measurements. Unless otherwise stated, the most recent data measured before the first dose of the study drug is used as baseline data.PK/PD analyses are performed with Phoenix WinNonlin software (version 8.1, Certara, Inc.).

## Ethics

The Institutional Review Board of Fukushima Medical University approved this trial on 22 February 2022 (IS03003). The Pharmaceuticals and Medical Devices Agency accepted an investigational new drug application for this trial on 16 March 2022. This trial was registered to jRCT on 17 June 2022 (jRCT2021220012). We are conducting this trial in accordance with the ICH Good Clinical Practice Guideline (ICH-GCP) and the Act on Securing Quality, Efficacy and Safety of Products Including Pharmaceuticals and Medical Devices of Japan.

## Quality management

Data management, Monitoring, and Audit are conducted by the contract research organization according to the ICH-GCP.

## Discussion

[211At] At is an α-emitting nuclide with a short half-life, and because α-rays have shorter range in water than β-rays, there is no need to isolate patients treated with this drug for long periods in an isotope treatment room. Possible side effects of [211At] At include myelosuppression, gastrointestinal toxicity, liver damage, and crisis because of catecholamine release, similar to those of other cytotoxic drugs.

[211At] MABG has been shown to have a cytotoxic effect on human medulloblastoma cells in vitro [12]. In an in vitro study using rat pheochromocytoma cell PC12, [211At] MABG dose-dependently increased the proportion of cells with DNA double-strand breaks and decreased cell viability [13,14]. [211At] MABG was administered to mice with subcutaneous transplantation of PC12 cells, and [211At] MABG showed high accumulation in PC12 cells and inhibited the proliferation of PC12 cells in a dose-dependent manner.

In a single-dose intravenous administration pharmacokinetic study in normal mice that we previously conducted (data not published), [211At] MABG was rapidly translocated from the blood into tissues, and the blood radioactivity concentration decayed biphasically with a $T_{1/2\alpha}$ half-life of about 11 minutes and a $T_{1/2\beta}$ half-life of 13.6 hours. After administration of [211At] MABG, radioactivity distribution in the blood was high in the blood cell fraction, and no free unchanged drug was detected in the plasma fluid fraction 60 min after administration. Excretion after administration of [211At] MABG showed a tendency similar to that of [123I] MIBG, but 6 hours after administration, [211At] MABG tended to be slower than [211At] MABG (49.3% ID, [123I] MIBG: 60.9% ID). Absorbed dose simulation of [131I] MIBG in humans using pharmacokinetics of [131I] MIBG in mice showed similar biodistributions to [131I] MIBG in humans [15–17], suggesting that mice could be appropriate test animals to predict absorbed dose of [211At] MABG in human.

The initial first human dose is recommended to be 1/10 of the rodent STD10 according to the ICH-S9 guideline. In our previous extended single-dose toxicity study of [211At] MABG in normal mice (BALB/c), three deaths out of 50 mice occurred in the 80 MB/kg dose group. However, no other irreversible toxicity was observed, and the severely toxic dose in 10% of mice (STD10) was considered to be ≥ 80 MBq/kg (data not published). Therefore, a dose of 8 MBq/kg in mice is used as the initial dose in the current clinical trial. When converted to a human-equivalent dose, this is 0.65 MBq/kg, so the common ratio is 1:2:4, and the dose levels are as follows: dose level 1: 0.65 MBq/kg, dose level 2: 1.3 MBq/kg, dose level 3: 2.6 MBq/kg.

We set the duration for DLT observation based on the findings of the above extended single-dose toxicity test (data not published). In mice, marked deterioration of general condition was observed from Day 5 to Day 13, and euthanasia and death were observed on Days 8 and 9 respectively. A blood test showed the lowest white blood cell count on Day 5, and it returned to normal thereafter. In the pathological examination on Day 35, the disorder of the small intestinal mucosa observed on Days 5 and 14 was not observed on Day 35. Therefore, we considered 35 days to be sufficient for determining DLT in humans, but we conservatively decided on an observation period of 42 days.

The investigational drugs used in the current study are manufactured in-house by Fukushima Medical University, and quality control is performed according to the Good Manufacturing Practice as much as possible. For safety, we wrote a manual on the proper use of [211At] MABG in clinical settings [19].

In the present study, up to 18 adult patients with PPGL are examined at a single center in Japan and receive a single bolus dose only. This is standard in first-in-human clinical trials and we consulted the Pharmaceuticals and Medical Devices Agency, before initiating this trial, and

[$^{211}$At] MABG was accepted as an investigational new drug application for clinical trial based on the results of our preclinical studies and the present study protocol.

Leveraging the results of the ongoing phase I trial, we are planning to conduct a phase II multicenter study with repeated recommended doses of [$^{211}$At] MABG for the patients with PPGL. In addition, [$^{211}$At] MABG might be a candidate for cancers which possibly uptake [$^{211}$At] MABG such as neuroblastoma.

## Conclusions

As the first low-molecular-weight compound labeled with an α-ray nuclide for human use in the world, we are accelerating the development of [$^{211}$At] MABG for the treatment of patients with PPGL.

## Supporting information

**S1 Checklist. SPIRIT 2013 checklist: Recommended items to address in a clinical trial protocol and related documents\*.**
(DOC)

**S1 File.**
(PDF)

**S2 File.**
(PDF)

## Acknowledgments

We thank Songji Zhao and Naoyuki Ukon for providing the results of preclinical studies necessary to conduct the clinical study.

## Author Contributions

**Conceptualization:** Masao Kobayakawa, Tohru Shiga.

**Data curation:** Tohru Shiga, Shigeyasu Sugawara.

**Funding acquisition:** Masao Kobayakawa.

**Investigation:** Masao Kobayakawa, Tohru Shiga, Kazuhiro Takahashi, Shigeyasu Sugawara, Kazuhiko Hanada.

**Methodology:** Masao Kobayakawa, Tohru Shiga, Kazuhiro Takahashi, Kaori Nomura, Kazuhiko Hanada, Naoki Ishizuka.

**Project administration:** Masao Kobayakawa.

**Supervision:** Naoki Ishizuka, Hiroshi Ito.

**Writing – original draft:** Masao Kobayakawa.

**Writing – review & editing:** Kaori Nomura.

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
