## [Decision Letter · Decision Letter 0]

12 Oct 2023

PONE-D-23-24585Evaluation of pharmacokinetics, safety, and efficacy of At-211 meta-astatobenzylguanidine (MABG) in patients with pheochromocytoma or paraganglioma (PPGL): A study protocolPLOS ONE

Dear Dr. Kobayakawa,

Thank you for submitting your manuscript to PLOS ONE. After careful consideration, we feel that it has merit but does not fully meet PLOS ONE’s publication criteria as it currently stands. Therefore, we invite you to submit a revised version of the manuscript that addresses the points raised during the review process.

We look forward to receiving your revised manuscript.

Kind regards,

Matteo Bauckneht

Academic Editor

PLOS ONE

Reviewers' comments:

Reviewer's Responses to Questions

**Comments to the Author**

1. Does the manuscript provide a valid rationale for the proposed study, with clearly identified and justified research questions?

Reviewer #1: Yes

2. Is the protocol technically sound and planned in a manner that will lead to a meaningful outcome and allow testing the stated hypotheses?

Reviewer #1: Yes

3. Is the methodology feasible and described in sufficient detail to allow the work to be replicable?

Reviewer #1: Yes

4. Have the authors described where all data underlying the findings will be made available when the study is complete?

Reviewer #1: Yes

5. Is the manuscript presented in an intelligible fashion and written in standard English?

Reviewer #1: Yes

6. Review Comments to the Author

You may also provide optional suggestions and comments to authors that they might find helpful in planning their study.

Reviewer #1: The work by Kobayakawa and colleagues illustrates the protocol of an ongoing study (started in September 2022) testing the efficacy of [211At]MABG as a therapeutic approach in adult patients (>20 years old) affected by pheochromocytoma/paraganglioma not amenable to surgery.

The study is interesting in light of the paucity of the currently available treatments in this setting and the potential to be transferred to other guanidine-avid diseases, such as neuroblastoma.

I suggest the following amendments to the text:

Please be consistent with the verbal tenses: over the manuscript, you switch from the present to the future tense repeatedly. Since the study is ongoing, you might want to stick to the present tense. Overall, the text reads fine; a review by a native English speaker might be, however, considered.

Please utilise the proper radiopharmaceuticals nomenclature, as indicated in the official EANM statement: https://www.eanm.org/content-eanm/uploads/2019/12/EANM_GUIDANCE-_TRACER_NOMENCLATURE-1.pdf

Please adjust the definition of “multiple active cancers”, which could be confusing in its current form; I suggest “presence of malignancies other than the one object of the study or history of other malignancies with a disease-free interval <5 years”

Please explain which drugs fall in the category described in the second point of the exclusion criteria; the recommended withdrawal period for these medications and the one mentioned in point 3 should also be added.

Please expand on the “uncontrolled thyroid dysfunction” that could lead to an exclusion from the study.

Discussion: consider changing the word “penetrability” to “shorter range in water”; the sentence “have a shorter half-life of At-211” referred to alpha-radiation is confusing and must be corrected or removed.

7. PLOS authors have the option to publish the peer review history of their article (what does this mean?). If published, this will include your full peer review and any attached files.

Reviewer #1: No

---

## [Author Response · Author response to Decision Letter 0]

7 Nov 2023

Responses to Reviewer’s

Comment 1. Does the manuscript provide a valid rationale for the proposed study, with clearly identified and justified research questions?

Response to Comment 1: Thank you for your valuable comment. This study is a first-in-human clinical trial of an anticancer drug. The rationale of the study is only from non-clinical pharmacological and biodistribution studies, which indicate the efficacy profiles for pheochromocytoma cells. They are described in the Introduction section (references: 12–17). We have added that the required toxicity studies were conducted prior to the first dose in humans, in accordance with ICH-M3 and -S9 guidelines (https://www.pmda.go.jp/files/000156321.pdf ). Based on these guidelines, we set the start dose for first administration in human, which is described in the Discussion section. 

Comment 2. Is the protocol technically sound and planned in a manner that will lead to a meaningful outcome and allow testing the stated hypotheses?

Response to Comment 2: This study was planned as a phase I study of an anticancer drug. The main purpose of a phase I cancer clinical trial is to determine the maximum tolerated dose and recommended dose by elucidating the dose-limiting toxicity in each dose escalation cohort. That is why, in the present study, there were no statistical hypotheses, and no statistical tests were planned. Regarding exploratory efficacy endpoints, such as objective response and progression-free survival, and pharmacological parameters, point estimations of these parameters are calculated.

Comment 3. Is the methodology feasible and described in sufficient detail to allow the work to be replicable?

Response to Comment 3: Sample size is not calculated statistically, but is traditionally set according to the modified Fibonacci dose escalation (3 + 3 design). In addition, there are usually no controls in phase I cancer clinical trials. Therefore, we have now added an explanation about the modified Fibonacci method in the section describing sample size and statistical analysis. We believe this manuscript ensures reproducibility of study planning and conducting.

Comment 4. Have the authors described where all data underlying the findings will be made available when the study is complete?

Response to Comment 4: We have added the following sentence to the data availability subsection in the METADATA: “No datasets were generated or analyzed during the current study. All relevant data from this study will be made available upon study completion.” 

 Comment 5. Is the manuscript presented in an intelligible fashion and written in standard English?

Response to Comment 5: We have checked this revised manuscript prior to resubmission, and had it proofread by native English-speaking scientific editors.

Comment 6. 

Comment 6-1: Please be consistent with the verbal tenses: over the manuscript, you switch from the present to the future tense repeatedly. Since the study is ongoing, you might want to stick to the present tense. Overall, the text reads fine; a review by a native English speaker might be, however, considered.

Response to Comment 6-1: Thank you for the advice, according to which we have amended the manuscript so that it now uses the correct tenses. In addition, as mentioned in our previous response, the manuscript has also been proofread by native-English-speaking scientific editors. 

Comment 6-2: Please utilize the proper radiopharmaceuticals nomenclature, as indicated in the official EANM statement: https://www.eanm.org/content-eanm/uploads/2019/12/EANM_GUIDANCE-_TRACER_NOMENCLATURE-1.pdf

Response to Comment 6-2: We have changed the radiopharmaceuticals’ nomenclature, in accordance with the EANM statement, from At-211 MABG, I-123 MIBG, and I-131 MIBG to [211 At] MABG, [123I] MIBG, [131I] MIBG, respectively.

Comment 6-3: Please adjust the definition of “multiple active cancers”, which could be confusing in its current form; I suggest “presence of malignancies other than the one object of the study or history of other malignancies with a disease-free interval <5 years”

Response to Comment 6-3: Thank you for your suggestion. We have now changed the sentence as follows: ““Multiple active cancers” refers to presence of malignancies other than the one included in the study or history of other malignancies with a disease-free interval <5 years.”

Comment 6-4: Please explain which drugs fall in the category described in the second point of the exclusion criteria; the recommended withdrawal period for these medications and the one mentioned in point 3 should also be added.

Response to Comment 6-4: We have now provided this information in the manuscript as follows: “Patients who cannot stop the dosage of a drug that suppresses the accumulation of MABG according to the EANM procedure guidelines for 131I-meta-iodobenzylguanidine (131I-mIBG) therapy during the study period.” In addition, we have clarified the withdrawal periods of 2 and 3 point as "during the study period."

Comment 6-5: Please expand on the “uncontrolled thyroid dysfunction” that could lead to an exclusion from the study.

Response to Comment 6-5: We have now changed “thyroid dysfunction” to “thyroid dysfunction including hypothyroidism and hyperthyroidism”.

Comment 6-6: Discussion: consider changing the word “penetrability” to “shorter range in water”; the sentence “have a shorter half-life of At-211” referred to alpha-radiation is confusing and must be corrected or removed.

Response to Comment 6-6: Accordingly, we have changed the word “penetrability” to “shorter range in water”, and revised the sentence as follows:

 “[211At] At is an α-emitting nuclide with short half-life, and because α-rays have shorter range in water than β-rays, there is no need to isolate patients treated with this drug for long periods in an isotope treatment room.”

---

## [Decision Letter · Decision Letter 1]

1 Feb 2024

PONE-D-23-24585R1Evaluation of pharmacokinetics, safety, and efficacy of [211At] meta-astatobenzylguanidine ([211At] MABG) in patients with pheochromocytoma or paraganglioma (PPGL): A study protocolPLOS ONE

Dear Dr. Kobayakawa,

Thank you for submitting your manuscript to PLOS ONE. After careful consideration, we feel that it has merit but does not fully meet PLOS ONE’s publication criteria as it currently stands. Therefore, we invite you to submit a revised version of the manuscript that addresses the points raised during the review process. Please submit your revised manuscript by Mar 17 2024 11:59PM. If you will need more time than this to complete your revisions, please reply to this message or contact the journal office at plosone@plos.org. Please include the following items when submitting your revised manuscript: A rebuttal letter that responds to each point raised by the academic editor and reviewer(s). You should upload this letter as a separate file labeled 'Response to Reviewers'.A marked-up copy of your manuscript that highlights changes made to the original version. You should upload this as a separate file labeled 'Revised Manuscript with Track Changes'.An unmarked version of your revised paper without tracked changes. You should upload this as a separate file labeled 'Manuscript'.If applicable, we recommend that you deposit your laboratory protocols in protocols.io to enhance the reproducibility of your results. Protocols.io assigns your protocol its own identifier (DOI) so that it can be cited independently in the future. For instructions see: https://journals.plos.org/plosone/s/submission-guidelines#loc-laboratory-protocols. Additionally, PLOS ONE offers an option for publishing peer-reviewed Lab Protocol articles, which describe protocols hosted on protocols.io. Read more information on sharing protocols at https://plos.org/protocols?utm_medium=editorial-email&utm_source=authorletters&utm_campaign=protocols.

We look forward to receiving your revised manuscript.

Kind regards,

Margo Dona

Academic Editor

PLOS ONE

Journal Requirements:

**Additional Editor Comments:** Please address the following points of reviewer 2:Include a comprehensive statistical analysis plan. Include details for determining the MTD and RD. Also, state the statistical methods that will be used to analyze the secondary endpoints. Identify the software that will be used to capture the data as well as the software that will be used for the statistical analysis.

Reviewers' comments:

Reviewer's Responses to Questions

**Comments to the Author**

1. Does the manuscript provide a valid rationale for the proposed study, with clearly identified and justified research questions?

Reviewer #1: Yes

Reviewer #2: Yes

2. Is the protocol technically sound and planned in a manner that will lead to a meaningful outcome and allow testing the stated hypotheses?

Reviewer #1: Yes

Reviewer #2: Yes

3. Is the methodology feasible and described in sufficient detail to allow the work to be replicable?

Reviewer #1: Yes

Reviewer #2: No

4. Have the authors described where all data underlying the findings will be made available when the study is complete?

Reviewer #1: Yes

Reviewer #2: No

5. Is the manuscript presented in an intelligible fashion and written in standard English?

Reviewer #1: Yes

Reviewer #2: Yes

6. Review Comments to the Author

You may also provide optional suggestions and comments to authors that they might find helpful in planning their study.

Reviewer #1: Thank you for addressing my comments. I have no further concerns against publication.

Reviewer #2: In this protocol, a phase I clinical trial is currently underway with 3 + 3 dose escalation design to evaluate the pharmacokinetics, safety, and efficacy of [211At] MABG at 3 dose levels in patients with unresectable or metastatic PPGI. The primary endpoint is dose-limiting toxicity to determine the maximum tolerated dose and recommended doses. The secondary endpoints include radiopharmacokinetics, urinary radioactive excretion rate, urinary catecholamine response rate, objective response rate, progression free survival, [123I] MIBG scintigraphy on reducing tumor accumulation, and quality of life.

Major revisions:

Include a comprehensive statistical analysis plan. Include details for determining the MTD and RD. Also, state the statistical methods that will be used to analyze the secondary endpoints.

Minor revisions:

1-Identify the software that will be used to capture the data as well as the software that will be used for the statistical analysis.

2-To assist in the review process, add line numbering to the document.

7. PLOS authors have the option to publish the peer review history of their article (what does this mean?). If published, this will include your full peer review and any attached files.

Reviewer #1: No

Reviewer #2: No

---

## [Author Response · Author response to Decision Letter 1]

25 Mar 2024

Responses to Reviewer’s

Comment of Reviewer 2. 

Major revisions: Include a comprehensive statistical analysis plan. Include details for determining the MTD and RD. Also, state the statistical methods that will be used to analyze the secondary endpoints.

Minor revisions. 1-Identify the software that will be used to capture the data as well as the software that will be used for the statistical analysis. 2-To assist in the review process, add line numbering to the document.

Response to all Comments of Reviewer 2: Thanks for your valuable comments. We have added following sentences to the “Statistical analysis” section (pages 20 to 21): 

“MTD is determined as the highest dose level at which none of the three cases developed DLT or only one of the six cases developed DLT. RD is determined as the same dose of MTD. All secondary endpoints are described as descriptive statistics, no formal statistical hypothesis test is performed.　Statistical analyses are performed with SAS software, version 9.4 (SAS Institute).” 

“PK/PD analyses are performed with Phoenix WinNonlin software (version 8.1, Certara, Inc.).”

---

## [Decision Letter · Decision Letter 2]

30 Apr 2024

Evaluation of pharmacokinetics, safety, and efficacy of [211At] meta-astatobenzylguanidine ([211At] MABG) in patients with pheochromocytoma or paraganglioma (PPGL): A study protocol

PONE-D-23-24585R2

Dear Dr. Kobayakawa,

We’re pleased to inform you that your manuscript has been judged scientifically suitable for publication and will be formally accepted for publication once it meets all outstanding technical requirements.

Kind regards,

Margo Dona

Academic Editor

PLOS ONE

Additional Editor Comments (optional):

Reviewers' comments:

Reviewer's Responses to Questions

**Comments to the Author**

1. Does the manuscript provide a valid rationale for the proposed study, with clearly identified and justified research questions?

Reviewer #2: Yes

2. Is the protocol technically sound and planned in a manner that will lead to a meaningful outcome and allow testing the stated hypotheses?

Reviewer #2: Yes

3. Is the methodology feasible and described in sufficient detail to allow the work to be replicable?

Reviewer #2: Yes

4. Have the authors described where all data underlying the findings will be made available when the study is complete?

Reviewer #2: No

5. Is the manuscript presented in an intelligible fashion and written in standard English?

Reviewer #2: Yes

6. Review Comments to the Author

You may also provide optional suggestions and comments to authors that they might find helpful in planning their study.

Reviewer #2: All comments have been adequately addressed.

7. PLOS authors have the option to publish the peer review history of their article (what does this mean?). If published, this will include your full peer review and any attached files.

Reviewer #2: No

---

## [Editor Report · Acceptance letter]

10 May 2024

PONE-D-23-24585R2 

PLOS ONE

Dear Dr. Kobayakawa, 

I'm pleased to inform you that your manuscript has been deemed suitable for publication in PLOS ONE. Congratulations! Your manuscript is now being handed over to our production team.

Kind regards, 

on behalf of

Dr. Margo Dona 

Academic Editor

PLOS ONE